# The Effect of Brick and Granite Block Paving Materials on Traffic Speed

**DOI:** 10.3390/ijerph16193704

**Published:** 2019-10-01

**Authors:** Xavier Rojas Nogueira, Jeremy Mennis

**Affiliations:** Department of Geography and Urban Studies, Temple University, Philadelphia, PA 19122, USA; jeremy.mennis@temple.edu

**Keywords:** paving materials, traffic calming, traffic speed, pedestrian safety

## Abstract

Slowing traffic speed in urban areas has been shown to reduce pedestrian injuries and fatalities due to automobile accidents. This research aims to measure how brick and granite block paving materials, which were widely used historically prior to the use of asphalt paving in many cities, may influence free flow traffic speed. Traffic speeds for 690 vehicles traversing street blocks paved with asphalt, granite block, and brick materials were measured using a radar gun on a sample of 18 matched pair (asphalt and historic paving material) street blocks in Philadelphia, Pennsylvania. Fixed effects linear regression was used to estimate the effect of paving material on vehicle speed after controlling for the street class (e.g., arterial versus local road) and the matched pair. Results indicate that brick reduced speeds by approximately 3 mph (~5 km/h) and granite block reduced speeds by approximately 7 mph (~11 km/h), as compared to asphalt paved city streets, which we attribute to drivers intentionally slowing due to road roughness. This research suggests that brick and granite block paving materials may be an effective traffic calming strategy, having implications for reducing negative health outcomes associated with pedestrian–automobile collisions.

## 1. Introduction

Injuries and fatalities due to motor vehicle accidents remain a serious public health issue across the globe. In the US, pedestrian fatalities due to collisions with vehicles have surged since 2009, increasing 11% from 2015 to 2016 alone [1]. Accidents are directly related to the speed at which traffic is moving, where higher vehicle speeds increase the danger that cars, trucks, and other vehicles pose to pedestrians, bicyclists, and other drivers [2,3]. In addition, fast moving traffic creates a sense of fear that erodes the quality of life of pedestrians, cyclists, and residents, particularly on residential and commercial streets.

A number of traffic calming approaches have been developed to slow vehicle speeds in cities, including street narrowing, speed bumps, and the insertion of traffic circles, planters and other obstacles along or in the roadway, among other approaches. One potential approach which has been given little attention is to utilize brick or granite block paving materials as a traffic calming technique, which has the added benefit of both increasing aesthetic quality and exploiting a resource that already commonly exists in many established urban areas. While brick and granite block paving materials are currently used in cities and towns across the world, in many cities in the US and elsewhere they often exist as historic artifacts, as for most cities, the use of brick and granite block paving materials was discontinued after the introduction of asphalt paving in the late 19th century. Even in these cities, however, brick and granite block paving surfaces often remain distributed piecemeal throughout the urban fabric, continuing to serve as part of the urban transportation infrastructure.

This research aims to quantify how two widely used historic urban street paving materials—brick and granite block (which we refer to as historic paving materials for shorthand hereafter)—may alter free flow traffic speed. Our motivation is that while historic paving materials may serve as an effective and aesthetic method to reduce vehicle speed, there has been relatively little research quantifying this effect. Understanding the impact of historic paving materials on vehicle speed would contribute to the potential development of the use of such materials for traffic calming in urban areas, enhancing pedestrian safety (particularly in older cities where historic paving materials may already be prevalent, or underlie current road surfaces paved with asphalt).

To this end, we measure and compare traffic speeds on asphalt, brick, and granite block paved streets in Philadelphia, Pennsylvania, a northeastern US city of approximately 1.6 million people. Philadelphia was founded in 1682 and has a planned rectangular street grid that covers the majority of the city. Although the vast majority of Philadelphia’s streets are now paved with asphalt, some streets maintain their original paving material [4] and possess significance as surviving fragments of the city’s history [5]. Philadelphia’s historic streets present a unique opportunity to test how historical paving materials affect traffic speed in cities, with important public health implications for their use as an urban traffic calming strategy.

## 2. Traffic Calming Strategies and Road Characteristics

Traffic calming, defined by the Institute of Traffic Engineers as ‘the combination of many physical measures that reduce the negative effects of motor vehicle use, alter driver behavior and improve conditions for non-motorized street users’ [6], is currently the most popular way to address speeding problems in both urban and rural areas. Here, traffic calming strategies are reviewed, with an emphasis on the effect of road geometry and pavement roughness on vehicle speed. Road geometry refers to the physical features of a segment of road such as (but not limited to): paving material, width, slope, appearance, and curvature [7], and can have pronounced effect on driver behavior, including the choice of route and free flow speed [8], defined by the 2010 Highway Capacity Manual [9] as ‘the theoretical speed on a study segment when density and flow rate are both zero,’ i.e., the speed a driver selects when other vehicles are not present.

Lane width and lateral clearance are both shown to have a considerable effect on driver behavior. Research indicates that a one-meter reduction in lane width is associated with a 3.5 miles per hour (mph) decrease in speed [10] and that placing obstacles adjacent to the road reduces speeds by an average of 13% [11]. In addition, although not strictly lateral clearance, the presence of homes and driveways around a road was shown to be associated with slower driving speeds [12]. Variables affecting visibility are an important determinant for driving speeds, including rain, sun glare, and road gradient, which when steep can obstruct driver vision [13]. 

Research indicates that road roughness plays an important role in reducing free flow vehicle speed, although the magnitude of the effect is variable [14]. Cooper et al. [15] observed an increase of 1.6 mph on roads following repaving. Te Velde [16] found an average speed reduction of 5% when road segments transitioned from smooth to rough surfaces, but found no immediate increase in speed when roads transitioned from rough to smooth surfaces. Yu and Lu [17] quantified the effect of surface roughness on speed as a decrease of 0.0083 mph for every one inch/mile increase in roughness. Chandra [18] found that such an effect may differ according to the type of vehicle, where a stronger effect was found for passenger cars as compared to trucks and other heavy vehicles. 

Not surprisingly, faster vehicle speeds are associated with an increased risk of accidents and more severe outcomes in pedestrian–vehicle collisions. The relationship between automobile related risk and speed is not linear, however, as accident rates increase at an increasing rate with incremental increases in vehicle speed. This suggests that even small reductions in speed can have substantial results on public safety. According to Taylor et al. [2], a 1 mph reduction in average speed is associated with a 6% reduction in accidents on low speed urban roads, 4% on medium speed roads, and 3% on high speed roads. In addition to automobile accidents, the probability of pedestrian fatality is highly sensitive to vehicle speed; Tefft’s [3] model estimates the probability of a struck pedestrian dying from a vehicle collision rising from 10% at 23 mph to 25% at 32 mph. At vehicle speeds between 32 and 50 mph, a 1 mph increase in speed is associated with a 2.8% higher chance of fatality. Other studies have found similar non-linear relationships between pedestrian risk and speed [19,20,21,22]. Consequently, vehicle speeds that appear to differ only marginally can pose substantially different risks to pedestrian safety.

Children are particularly vulnerable to the dangers of automobiles, especially in residential areas, with collision involvement rates in the US being the highest for 5–9-year-old males [23]. The perception of streets as dangerous due to traffic can also have secondary detrimental effects on neighborhood life, particularly for children, since it strips the street of its ability to be used for outdoor recreation and socialization [24].

Traffic calming addresses the risks associated with high vehicle speeds as well as reduces noise, encouraging ‘green’ modes of transportation like cycling, and makes a given street feel safer. The first purposeful traffic calming strategies in the US were implemented in the 1970s. Although traffic calming is intended to be integrated with driver culture and cultural norms of transportation behavior [25], devices used to force or facilitate reductions in speed remain the primary mode of traffic calming. Speed bumps remain the most popular traffic calming device in the United States [26] due to their high effectiveness relative to their cost. Speed bumps have been shown to reduce child injury rates, making children safer in their neighborhoods [27], by forcing drivers to reduce vehicle speed to about 15–20 mph in order to avoid discomfort [28].

Speed bumps may, however, increase speeds on adjacent streets. Werner [29] reported a 23% increase in speeding on a street adjacent to a street with installed speed bumps. In addition, speed bumps cause discomfort to bicyclists traveling at lower speeds [26]. Chokers, which are areas of road narrowing via curb extension, are another approach to speed reduction [6]. Chokers have demonstrated a statistically significant speed reduction effect, but the effect dissipates after the vehicle exits the choker. Chokers also cause cyclists to feel less safe due to the sudden narrowing of the street [30]. Reconfiguring a four-lane to a three-lane road also appears to reduce speeds, but only after a critical traffic volume. Similar to speed bumps, raised sidewalks are associated with highly variable driver behavior responses, and do not significantly reduce approach or exit speeds.

To our knowledge, there is limited literature quantifying the effect of alternative or historic paving materials on free flow vehicle speed. A 1984 study found a significant reduction in speed on a road that transitioned from asphalt to brick [31], and Van de Kerkhof [12] found similar results comparing a few brick and asphalt streets. A more recent Japanese study [32] looking at brick versus red-painted crosswalks found that ‘brick and red-colored pavements are equally effective for reducing the driver’s speed, though, for right-turning vehicles, the brick pavement is more effective for reducing the car speed at the crosswalk at a signalized intersection’. The present research aims to contribute to a better understanding of how brick and historic paving materials influence vehicle speed, which can assist planners in incorporating such paving materials into traffic calming strategies as a part of comprehensive transportation planning.

## 3. Philadelphia’s Brick and Granite Block Paved Streets 

As with many cities in the US, Philadelphia utilized granite block and brick paving from the mid-1800s until the dawn of cheap asphalt in the late 1920s. Prior to American independence in 1776, streets were primarily composed of dirt and gravel, though the health benefits of paving streets were known [5]. By 1830, most Philadelphia streets were paved with cobblestone [33]. In 1891, civil engineers began advocating for modern paving materials for different circumstances. High traffic areas within the city were recommended to use granite block while brick, the smoother alternative, was advocated for lower traffic areas and was used in the suburbs and more peripheral areas of the city. By 1916, only 0.3% of Philadelphia streets remained cobbled while the remainder used the new granite block and brick paving materials. In the early 1900s, street paving strategies shifted to support automobile transportation. Asphalt, which became affordable once chemists were able to produce it artificially in America, provides a smooth surface for automobile transportation and was also considered easy to clean and thus more sanitary than other materials [34]. By 1920, the city had paved over the majority of the historic cobblestone, brick, and granite block paving materials with asphalt.

Certain street blocks in Philadelphia were never paved over with asphalt for various reasons, yet they slowly decayed and degraded, maintained occasionally with asphalt ‘plumber patches.’ This continued until the end of 1998, when the Historic Street Paving Thematic District was created by the City of Philadelphia, which sought to preserve the ‘collection of several hundred blocks of streets in the city that retain their historic street paving materials’ [4]. Presently in Philadelphia, 328 blocks on 159 streets remain paved with their historic materials [35], primarily brick (Figure 1) or granite block (Figure 2). While Philadelphia’s historic paving materials are protected due to their historical significance today, they have not been considered for their use as traffic calming devices.

## 4. Data and Methods

### 4.1. Streets Data

Data on the addresses of the Philadelphia street blocks paved with brick and granite block were acquired from the Philadelphia Historic Street Paving Thematic District. The addresses were last updated in 2016, and for each address, the paving material and historic integrity were listed. To verify the accuracy of the list and to ensure that addresses were not re-paved, Google Maps Street View (Google LLC, Mountain View, CA, USA) and in-person site visits were used. The Historic Paving District categorizes all street segments as either low, medium, or high quality, referring to the physical integrity of the historic paving material. After utilizing Google Street View to view all the low-quality segments, they were excluded from the analysis due to the fact that many of them were extremely damaged uneven road surfaces, often with very little of the historic material remaining on the road. Also excluded were dead-end streets and extremely narrow alleys, as these do not generally experience free flow traffic, as well as streets that are only partially covered by historic paving materials. Other reasons for excluding a street block from consideration included the presence of construction or other traffic calming devices, such as speed bumps or rumble strips. Blocks with extensive on-street parking, which disrupts free flow traffic as drivers search for parking spots, or blocks with very little traffic to observe, were also excluded. Street blocks used in the study were generally standard block lengths for Philadelphia (400–500 feet), straight, with mid-block access limited to small alleys with very little traffic. 

### 4.2. Matching Historically Paved and Asphalt Street Blocks for Comparison

We employ a matched pair analytical design where we compare vehicle speeds between two matched street blocks, one of which is paved with historic paving materials and one of which is paved with asphalt, but which are otherwise similar in regards to other characteristics that are known to influence vehicle speed. Such a design allows for the control of extraneous variables that may confound estimates of the effect of paving material on vehicle speed. Historic paving material streets for this study were selected via random sample. Of the street blocks eligible for analysis, the inventory was split into two tables consisting of granite block paved street blocks and brick paved street blocks. For each table, a random number generator was used to select ten granite block paved street blocks and eight brick paved street blocks for a total of 18 sampled blocks. Each of these street blocks paved with historic materials was then matched to a similar, matching street block paved with asphalt for comparison of vehicle speed (Table 1), using the following strategy adapted from other geographic approaches to matched pair analytical designs [36].

The ArcGIS Geographic Information System (GIS) software package (ESRI, Inc., Redlands, CA, USA) was used to map the historically paved street blocks, along with the asphalt paved street blocks, through a joint operation with the Philadelphia Streets Centerline spatial data layer acquired from the City of Philadelphia Planning Department. If an adjacent block of the same street was paved in asphalt rather than a historic material—i.e., where drivers continued on the same street from a block with an asphalt to a historically paved surface or vice versa—it was used as a match. Otherwise, the asphalt street block nearest to the historically paved street with the same characteristics, including direction of traffic, speed limit, width, and Philadelphia Streets Department classification (major arterial, minor arterial, collector, or local street) as the historically paved street, was chosen as a match. In addition, Google Street View and site visits were used to evaluate other characteristics that could influence the appropriateness of a given match, including street gradient (slope), street width, parking patterns, and pedestrian activity. If one of these characteristics of the nearest asphalt street block of the same class prohibited its role as a match, the next nearest candidate asphalt paved street block was considered. Of the 18 matched pairs of street blocks, seven are local streets, seven are collector streets, three are minor arterial streets, and one pair (adjacent blocks of Chestnut Street) is a major arterial street.

### 4.3. Capturing Traffic Speed

The speed of individual vehicles on each street block was recorded using a Pocket Radar (Pocket Radar Inc., Santa Rosa, CA, USA) ‘Personal Speed Radar’ device by an observer standing at the end of the street block recording vehicle speed continuously as each vehicle drove down the length of the street in either direction. The observer stood in between parked cars or sat on the side of the road and aimed the Pocket Radar device without outstretching the arm in order not to distract drivers and thus affect vehicle speed while taking speed measurements. Recordings were taken at as close to parallel to the movement of the vehicle as possible, and the maximum observed speed for each vehicle was recorded. 

Because many of the streets do not carry substantial traffic volume, all recording where taken during the morning (7 a.m.–9 a.m.) or afternoon (4 p.m.–6 p.m.), which ensured a substantial number of vehicles to be measured. Sampling occurred in dry conditions, and recordings were taken within one hour for each matched asphalt/historic paving surface sample pair of street blocks. Vehicles traveling in dense traffic were not measured—only vehicles which appeared to travel without hindrance or driver distraction down the length of the street block were sampled. Certain types of vehicles were excluded from the sampling, including buses and large or commercial trucks, emergency vehicles, motorcycles and three-wheeled motorized vehicles. Vehicles were excluded from the sample if there were disruptions to free flow traffic, including the presence of pedestrians crossing or standing in the road, cars pulling out of parking spots, stopped cars in the road, backed-up traffic at a stop light or stop sign, or if something of interest was happening on the side of the road which caused drivers to divert their interest. 

### 4.4. Analytic Plan

This analysis aims to estimate the effect of brick and granite block historic paving materials on vehicle speed while controlling for other factors, such as the street class. We first generate descriptive statistics for vehicle speeds for each paving material (brick, granite block, and asphalt). We also note that the sampling is clustered, with 690 observations ascribed to 18 matched pairs of street blocks. The intraclass correlation coefficient (ICC) is 0.214, which indicates that a relatively low proportion of variance in vehicle speed is explained by between-cluster effects. In other words, vehicle speed does not vary substantially from one set of matched pair of street blocks to another. Nonetheless, we acknowledge that controlling for variation across matched pairs may improve our estimation of the effect of paving material on vehicle speed. Thus, to address within-matched pair clustering, we employ fixed effects linear regression (which has been found to be robust to smaller cluster and within-cluster sample sizes [37]) to estimate the effect of paving material on vehicle speed while controlling for street class and the matched street block from which the observations were collected. 

We begin by estimating the unadjusted effect of paving material on speed, where asphalt serves as the referent category. In a second model, we estimate the adjusted effect of paving material by adding street class as a control variable, where ‘major arterial’ serves as the referent category. In a third model, we estimate the adjusted effects of the paving material on vehicle speed following fixed effects modeling of the matched pair of street blocks by adding a set of dummy variables that uniquely identify the street block pair, i.e. a unique identifier for each historically paved street block and its asphalt paved matching street block, expressed as:(1)$$S_{ib}=P_{b}+C_{b}+D_{b}+\epsilon_{i}$$
where $S_{ib}$ is the estimated speed of vehicle *i* on matched pair street block *b*, P is the pavement type (brick or granite block categories, with asphalt serving as the referent category), C is the street class (minor arterial, collector, or local categories, with major arterial serving as the referent category), D is a vector of dummy variables indicating the matched pair of street blocks, and ϵ is the error term. This fixed effects approach controls for unaccounted differences among matched pairs of street blocks, thus allowing us to isolate the effect of paving material on vehicle speed. We then use a stratified design to investigate whether the effect of paving material on vehicle speed differs by street class by running a separate fixed effects regression model on the set of observations within each street class category (minor arterial, collector, and local streets), excluding the major arterial category as there is only one matched pair with 38 total vehicle speed observations.

## 5. Results

A total of 690 vehicle speed observations were made for 18 matched pairs of historically paved and asphalt street blocks. The number of vehicle speed observations per matched pair of street blocks ranged from 28–40, with 341 (50%), 153 (22%), and 196 (28%) observations occurring for asphalt, brick and granite block, respectively. The number of observations taken for major arterial, minor arterial, collector, and local street blocks was 38 (6%), 108 (16%), 279 (40%), and 265 (38%), respectively. Table 2 shows the descriptive statistics of observed vehicle speeds for asphalt, brick, and granite block paving materials, with asphalt having the highest average speed and granite block having the lowest, as expected. Histograms indicate that the vehicle speeds for the different categories of paving materials do not depart substantially in shape from a normal distribution.

Results of the unadjusted, adjusted, and adjusted fixed effects linear regressions of vehicle speed are reported in Table 3. The unadjusted model (Model 1) indicates that brick and granite block lowers vehicle speed by approximately 2.5 mph and 7 mph, as compared to asphalt, with 28% of the variation in vehicle speed explained by paving material alone.

Forty-five percent of the variation in vehicle speed is explained by Model 3. Including other variables that describe street characteristics that could affect vehicle speed may increase the explanatory power of the model, though we note that the fixed effects accounts for individual variation among the matched pairs of street blocks. Thus, we ascribe most of the remaining variation in vehicle speed to differences in behavior among individual drivers.

Table 4 reports the results of the fixed effects regression stratified by street class. Results indicate that the effect of paving material is least for minor arterial streets and greatest for local streets. For minor arterial streets (Model 4), the effect of brick on vehicle speed is not significant (though the effect of granite block is significant), suggesting the effect of brick on vehicle speed is driven primarily by collector and local streets. The effect of granite block on reducing vehicle speed is particularly strong for local streets (Model 6) as compared to the other street classes, where over 50% of the variation in vehicle speed is explained by this model.

## 6. Discussion 

This research contributes to a better understanding of how urban historic street paving surfaces, specifically brick and granite block, impact vehicle speeds. Our results suggest that both brick and granite block paving materials indeed reduce urban free flow vehicle speed. We find that brick reduces speeds by approximately 3 mph (~5 km/h) and granite block reduces speeds by approximately 7 mph (~11 km/h), which represents a reduction of approximately 13% and 31%, respectively, from the average speed we observed on asphalt paved city streets in Philadelphia. 

We ascribe this reduction in speed for streets paved with historic paving materials to the increase in road roughness associated with brick and granite block road surface as compared to asphalt. Our results are consistent with previous research which shows that road roughness is associated with slower vehicle speeds [15,16,17,18]. Rougher road surfaces can cause drivers to slow down due to the physical discomfort of driving over uneven road surfaces [38]. Other researchers have suggested that drivers may also slow down due to the attention demanded to navigate over rougher road surfaces [31,39,40].

While reductions of 3 mph and 7 mph may not appear to be large changes in absolute speed, research has demonstrated that even marginal changes in vehicle speed can have substantial impacts in the rate of automobile-related pedestrian fatalities. In an analysis of 1994–1998 US vehicle–pedestrian crash data, Tefft [3] assessed the association between vehicle impact speed and the risk of severe injury, as well as the risk of death. Not only does the risk of injury and death increase with higher vehicle speeds, the association is non-linear at speeds below 32 mph. Thus, even modest reductions in vehicle speed are likely to result in substantial decreases in negative pedestrian health outcomes. In 2016–2017 alone, 81 pedestrians and bicyclists were killed by vehicle collisions in Philadelphia [41]. Reducing vehicle speeds by even modest amounts through traffic calming strategies is thus a key objective in making Philadelphia’s, and other cities’, streets safer for pedestrians, bicyclists, and other drivers [42]. 

Interestingly, we found that the mitigating effects of historic paving materials on vehicle speed were particularly acute for local roads, with arterial and collector streets having lesser effects, particularly for granite block paved street surfaces. We speculate this may be because of the greater traffic volume on streets classified as arterial and collector roads, as compared to local roads. Although we took care to measure free flow traffic and eliminate observations of vehicles whose speeds were clearly influenced by the presence of other vehicles, parking, visual distractions, or other influences outside of the paving material, it may be that drivers on these busier arterial and collector streets are influenced by the presence of other more distal vehicles; or perhaps merely the role of the street itself as an arterial, even when in free flow traffic conditions, may dampen the effect of the paving material on vehicle speed. We note that the matched pair of street blocks classified as major arterial streets passed through Independence National Historical Park, the historic tourist district of the city, and it is possible that general park activity may have caused drivers to drive more slowly relative to drivers on other street blocks in the sample. 

We acknowledge several limitations to the study. First, though we were careful to exclude drivers clearly looking for parking or slowing for other reasons unrelated to the paving surface, it is possible that not all vehicles in the study were in ideal free flow traffic conditions. In addition, we could not control for drivers choosing which street block to traverse. Speed recordings may also be subject to recorder and observer error. A greater number of individual vehicle speed recordings would confer greater confidence to our results. The research design may be improved by having matched pairs of street blocks with different paving materials adjacent to one another, such that the same driver stays on the same street and merely changes road surfaces. Or, alternatively, one would benefit from a natural experiment where a historically paved street block is paved over with asphalt, and thus one would be able to measure vehicle speeds before and after the repaving. However, neither of these options were available for the present analysis. We also acknowledge that the fixed effects model represents only one approach to address clustering in the sample. We repeated the analysis using generalized estimating equations (GEE) as another approach to treating the within-matched pairs clustering; results regarding the effects of historic paving materials on vehicle speed were substantially the same as the fixed effects models. 

We note that while our findings may be extended logically to other cities in the northeast US with relatively similar street infrastructure and paving histories, such as Boston, New York, and Washington, D.C., the generalizability of our results to other cities in the US where road infrastructure is newer (i.e., streets are wider and smoother), such as in Los Angeles or Phoenix, and drivers may not be as familiar with driving over historic paving materials, is unknown. Drivers in cities in other countries may also be more (or less) comfortable with non-asphalt paved or rougher road surfaces than drivers in Philadelphia and thus may behave differently.

## 7. Conclusions

Our results suggest that historic paving materials on urban streets are associated with reduced vehicle speeds, as well as provide some measure of the magnitude of speed reductions for certain historic paving materials among different street classes (e.g., local streets). Consequently, this research suggests that historic paving materials can be used strategically for traffic calming, and can serve a similar purpose as the more commonly used speed bumps, rumble strips, chicanes, and the like. From a public health and planning perspective, cities may choose to employ historic paving materials to slow traffic as a means to reduce pedestrian and bicyclist injuries and fatalities due to automobile accidents. Notably, historic paving materials such as brick and granite block are considered aesthetically pleasing, and can serve a dual purpose of traffic calming and beautification, and thus may be a more attractive alternative than other traffic calming devices used to enhance pedestrian safety. In many older cities, such as Philadelphia, historic paving materials were often never removed and were merely paved over with a layer of asphalt; thus, in many cases, they do not need to be newly installed, but rather revealed from the overlying asphalt. 

Further research in other US cities, as well as cities outside the US, would clarify the generalizability of our findings. It would also be useful, given the results found here, to compare the effect of historic paving materials on vehicle speed to other traffic calming measures, such as speed bumps or rumble strips, as well as to compare the economic, aesthetic, noise, and vehicle impacts of these strategies, so that policy makers can evaluate the efficiency of using historic paving materials as a traffic calming device. Research on the durability of historic paving materials, as well as the potential effects on tire traction and vehicle braking speed, would be required to adequately assess the feasibility of introducing brick and granite block paving materials for traffic calming. Finally, it may also be useful to quantify the roughness effect of certain historic paving materials using a standardized measure such as International Roughness Index in order to compare results with previous research on the effect of asphalt roughness on vehicle speed [31,38,39,40].

## Figures and Tables

**Figure 1 ijerph-16-03704-f001:**
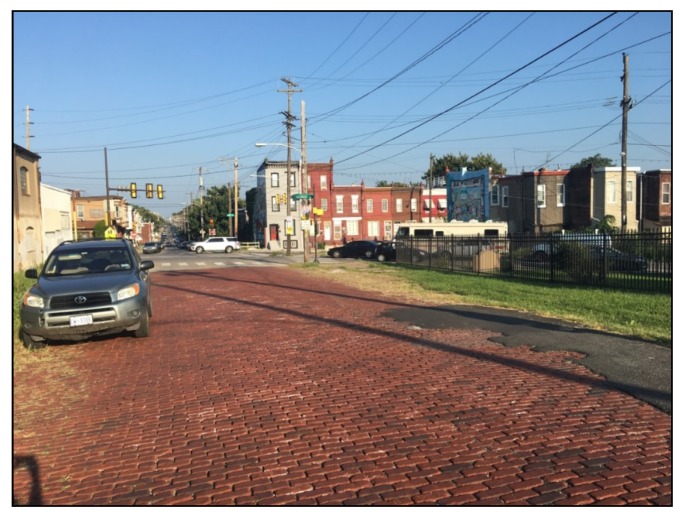
A Philadelphia street paved with brick historic paving material, located at 2900 W. Montgomery Ave.

**Figure 2 ijerph-16-03704-f002:**
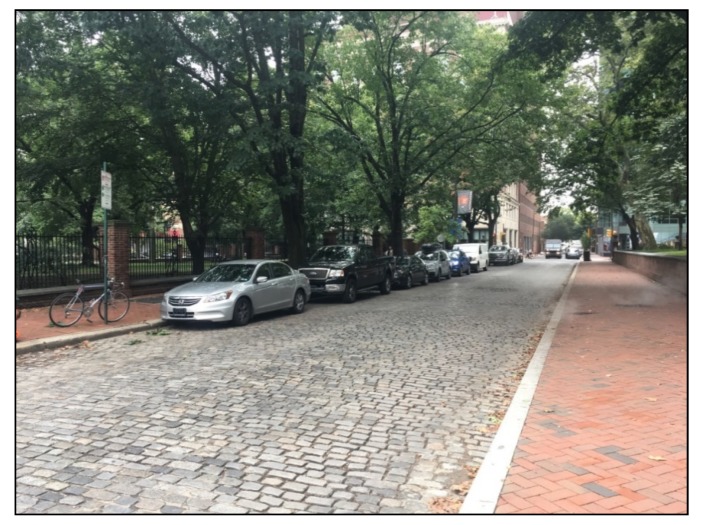
A Philadelphia street paved with granite block historic paving material, located at 102 S. Independence Mall E.

**Table 1 ijerph-16-03704-t001:** List of historically paved street blocks and their asphalt paved match.

Historic Material Paved Street Block	Asphalt Matched Street Block	Street Class	Speed Limit ^1^
102 S. Independence Mall E. (Granite block)	200 S. 5th St.	Minor Arterial	25 mph
400 Front St. (Granite block)	1000 Germantown Ave.	Local	25 mph *
300 S. 2nd St. (Granite block)	700 S. 2nd St.	Minor Arterial	25 mph
100 Chestnut St. (Granite block)	300 Chestnut St.	Major Arterial	35 mph
4200 Station St. (Granite block)	4100 Tower St.	Local	25 mph **
4700 Smick St. (Granite block)	300 Conarroe St.	Local	25 mph **
4300 Cresson St. (Granite block)	4100 Cresson St.	Collector	25 mph **
2600 Trenton Ave. (Granite block)	2100 E Huntingdon St.	Collector	25 mph **
100 N. Croskey St. (Granite block)	300 S. 24th St.	Local	25 mph
250 Hermitage St. (Granite block)	350 Hermitage St.	Local	25 mph
100 W Sylvania St. (Brick)	100 Apsley St.	Local	25 mph **
300 Dupont St. (Brick)	450 Krams Ave.	Local	25 mph **
4520 Pechin St. (Brick)	4500 Mitchell St.	Collector	25 mph **
2900 W. Montgomery Ave. (Brick)	3000 Sedgley St.	Collector	25 mph
1000 S. 49th St. (Brick)	1200 S. 49th St.	Minor Arterial	30 mph
100 W. Abbottsford St. (Brick)	200 W. Abbottsford St.	Collector	25 mph **
700 Sansom St. (Brick)	900 Filbert St.	Collector	25 mph **
3426 W. Coulter St. (Brick)	3400 W. Coulter St.	Collector	25 mph **

^1^ Speed limit as posted; if no posted speed limit then *derived from Pennsylvania Department of Transportation, or **derived from the normative 25 mph speed limit which applies to Philadelphia residential streets without posted speed limits.

**Table 2 ijerph-16-03704-t002:** Descriptive statistics for vehicle speeds on different paving surfaces. SD = standard deviation.

Paving Material	Min (mph)	Max (mph)	Mean (mph)	SD (mph)
Asphalt	11	41	22.3	4.6
Brick	8	31	19.8	4.9
Granite Block	7	28	15.4	4.6

**Table 3 ijerph-16-03704-t003:** Results of linear regression of vehicle speed (unstandardized coefficients are reported with standard errors in parentheses; *** *p* < 0.005).

Variable	Variable Value	Model 1 (Unadjusted)	Model 2 (Adjusted)	Model 3 (Adjusted with Fixed Effects)
Constant		22.31 (0.25) ***	20.05 (0.76) ***	20.09 (0.70) ***
Block Material (Reference = Asphalt)	Brick	−2.57 (0.45) ***	−3.04 (0.45) ***	−2.93 (0.48) ***
	Granite Block	−6.92 (0.42) ***	−6.52 (0.41) ***	−6.59 (0.41) ***
Road Class (Reference = Major Arterial)	Minor Arterial		3.22 (0.86) ***	5.24 (0.93) ***
	Collector		3.32 (0.80) ***	3.51 (0.93) ***
	Local		1.06 (0.79)	1.54 (0.93)
Adjusted *R*^2^		0.28	0.33	0.45

**Table 4 ijerph-16-03704-t004:** Results of stratified fixed effects regression of vehicle speed by street class (unstandardized coefficients are reported with standard errors in parentheses; *** *p* < 0.005).

Variable	Variable Value	Model 4 (Minor Arterial, *n* = 108)	Model 5 (Collector, *n* = 279)	Model 6 (Local, *n* = 265)
Constant		24.40 (0.65) ***	18.61 (0.79) ***	17.20 (0.66) ***
Block Material (REF = Asphalt)	Brick	−1.14 (1.27)	−3.07 (0.64) ***	−3.27 (0.95) ***
	Granite Block	−4.75 (0.75) ***	−4.70 (1.01) ***	−7.29 (0.55) ***
Adjusted *R*^2^		0.47	0.25	0.51

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
