# Peer review of "The Effect of Brick and Granite Block Paving Materials on Traffic Speed"

_ijerph, 2019, doi:10.3390/ijerph16193704_

Round 1

Reviewer 1 Report

For the sample streets, more detailed information should be provided, including the length, traffic volume, number of accesses, etc. Using Poket Radar gun is not a good way to collect speed data, cause it’s not easy to eliminate the influence of observers and the system errors. Even so, as mentioned in the paper, “…standing at the end of the street block recording vehicle speed continuously as each vehicle drove down the length of the street in either direction”. The observer can collect multiple speed data, how did you determine the speed on the sample street? Random one or average? For the average speed collection, maybe the observers can track target vehicle through video recording the plate number at the start and end of sample street. Another research scheme is to analyze the speed trajectory data using filed naturalistic driving study. Total 690 vehicles for 18 matches pairs. The small data sample is weak to deduce solid conclusions. Wrong organization of tables 2-4. For the modelling results, multiple factors affect the running speed on street, not only the paving materials. Although the author declared that other factors had been controlled, I did not find any detailed data proving except qualitative description. Maybe historic paving materials have positive effect on traffic speed, it should be mentioned whether it can be recommended as a common traffic calming countermeasure. Maybe other factors could not be ignored, such as noise, construction and maintainance cost, etc.

Reviewer 2 Report

The research investigates the impact of paving material on vehicle speed.

The methodology is sound and the conclusions properly supported.

Indeed speed reduction in urban areas have shown substantial reductions in the rate of vehicle-related pedestrian fatalities. However, the authors should stress the fact that the impact from noise as well as tire wear during vehicle motion on the examined surfaces have not been examined.

Reviewer 3 Report

Review

Paper: The Effect of Historic Paving Materials on Traffic Speed

Comments: -

This type of pavement types are not historical; although they have been there before asphalt pavements are invented. They are still in use in many cities around the globe for the same purpose. So use another terminology .. like ‘A comparison study of different pavement types/materials for reduction of vehicle speed’. some language errors, example line 220 it should be ‘were’ instead of ‘where’ avoid using ‘our’, ‘we’ in articles … lines 233, 234 for example Explanation of the Fixed effects linear regression method is missing, and why it was chosen for this analysis. Provide the data range for the variables, with scatter plots if possible. Table 2 and Table 3, did they change place? Can the regression model equations be provided? With R2 as low as 0.25 – 0.51, the models explain none of the independent variables; hence, conclusions made should be checked again?

Round 2

Reviewer 1 Report

Some necessary explanation had been provided in the revised version.